# What Do Coordinators Do? Mental Health Policy Implementation as Translation

**Coralie Darcis** [1,]*  **and Sophie Thunus** [2]

1    Centre for Sociological Research and Interventions, Faculty of Social Sciences, University of Liège, 4000 Liège, Belgium

2    Public Health Faculty, Catholic University of Louvain, 1200 Wolluwé-Saint-Lambert, Belgium; sophie.thunus@uclouvain.be

*    Correspondence: coralie.darcis@uliege.be

**Abstract:** Coordination is described as a widespread function emerging in relation to policy plans inducing collaboration between different sectors, organizations and professions. This paper suggests seeing the implementation phase as a translation process, one where the content of policy plans is reinvented primarily through discussion rather than linearly transferred from the political to the professional arena. It focuses on the function of coordinator with a view to examining how this function is performed and questions its influence on the local translation of both policy plans. The data collection was part of two research projects focusing on the reform of Belgian mental healthcare and the creation of care pathways for forensic patients, combining document analysis, interviews (n = 82) and observations (n = 58). The results highlight the inherent ambiguity of the coordinators' working environment, the socially-disputed nature of their function and define the coordinators as connection-makers who exert power over processes rather than people or structures. It demonstrates that coordinators influence the policy process by inducing discussions at meetings and the documents subsequently produced. In conclusion, this paper defines coordinators as process managers whose work largely consists of translating policy plans through *event connectivity* and *contextualizing* practices. Given the importance of translation in policy implementation, this paper calls for a reconsideration of policy evaluation as well as of the coordinators' recruitment and training procedures.

**Keywords:** coordination; policy implementation; process management; improvisation; meetings and documents; mental healthcare policies; forensic policies

---

## 1. Introduction

This paper discusses the emergence of the function of coordination in relation with two policy plans, which are part of the same policy process aiming to reintegrate people with mental health problems into the community. It describes coordination as a widespread function arising in relation to policy plans targeting complex policy issues and thus entailing increased collaboration between different sectors, organizations and professions.

Policy makers approached the implementation as a creative process enabling adjustment to local needs and existing care services. Network coordinators were made responsible for managing this process through setting up local committees in charge of putting the care models outlined in the plans into practice.

This paper suggests seeing the implementation phase as a translation process, one where the content of policy plans is reinvented primarily through discussion rather than linearly transferred from the political to the professional arena. It focuses on the function of coordinator with a view to

ask the question "what do coordinators do?" and questions its influence on local translation of both policy plans.

In order to fully understand the function of coordination, this paper first addresses the political vision of the function and then moves on to theoretical and empirical accounts. The political vision will first provide information on the coordinators' skills and on the tasks they have to complete. Secondly, the scientific literature on coordination of health services emphasizes the uncertain environment in which coordination is enacted and defines this function as mainly consisting of boundary spanning and knowledge brokering activities. Thirdly, our empirical description aims to deepen our understanding of the work of coordinators by setting both the political vision and theoretical definition against the coordinators' perception of their work as well as against practical forms of coordination.

As a result, this paper identifies two core practices of coordination: meetings and interpreting and writing documents. Based on a close examination of how coordinators lead discussions in meetings and how they interpret documents, it highlights two methods where coordinators influence the translation of policy plans. The first is referred to as event connectivity and the second consists of a joint operation of selection and articulation—which we refer to as "contextualizing" the speeches of the participants at the meetings.

Finally, since coordinators use these methods mainly at meetings—the course of which is never fully predictable—this paper argues that the work of coordinators is one of improvisation. This concept of coordination as improvisation enables us to derive three main lessons concerning coordination and its influence on the policy process. These are discussed at the end of this paper.

## 2. Context: the Emergence of the Function of Coordinator

This paper focuses on two policy plans, which are part of the same policy process. This process follows an international trend towards deinstitutionalization, defined as a progressive shift from hospital-based to community mental health care (Novella 2010a, 2010b). In Belgium this process started in 2010 with the launch of Innopsy107, a policy plan designed to improve mental health for the general population. From 2012 onwards, Innopsy107 turned into a general framework into which other policy plans more specifically directed towards sub-groups of the population, such as forensic patients, were gradually incorporated.

This section describes the emergence of the function of network coordinator through the creation of Innopsy107 and the Forensic Masterplan. It draws attention to the challenges facing the function of coordinator and then outlines the political vision of this function.

### 2.1. Innopsy107

In 2010, the launch of Innopsy107 represented a milestone. This policy plan explicitly states its ambition to induce a shift from hospital to community mental health care. The World Health Organization (WHO) played a significant role in placing the improvement of community mental health care higher on the policy agenda in Belgium. In 2005 the Belgian public health ministers attended the WHO Inter-Ministerial Conference in Helsinki and subsequently decided to explore policy solutions in order to achieve this shift from hospital to community mental health care. Moreover, in 2008 a WHO report (World Health Organization 2008) demonstrated that the total number of psychiatric beds in Belgium was still too high in comparison to other European countries. This provided change advocates with arguments to convince Belgian policy makers to support a more general shift toward community psychiatry. In 2009, policy makers decided to launch a new mental health policy. A think tank composed of public officials and national experts was created to draw up a policy plan. The think tank's members had good knowledge of recent developments in the field of mental health, both nationally and internationally (Thunus 2015), (Thunus and Schoenaers 2017).

The think tank authored a policy plan outlining a new mental health care model as well as its implementation scheme (Service Public Fédéral Affaires Sociales et Santé Publique 2010). As described in the policy plan, the new model is community-based, centered on the patient's needs and covers

five care functions including mental health prevention and promotion, mobile psychiatric teams for acute and chronic diseases, rehabilitation facilities, intensive psychiatric treatments and alternative residential facilities. Implementing these five functions requires the development of local mental health service networks. A network coordinator is responsible for the development of each network, setting up and managing working groups known as "network and function committees". In the five years following the presentation of Innopsy107, twenty-one local networks developed throughout the country.

## 2.2. The Forensic Masterplan

The Belgian forensic system has faced a structural lack of places for forensic patients for several decades. Indeed, a large part of them were housed on prison psychiatric wards in unsuitable conditions. This situation was regularly condemned by organizations such as the European Committee for the Prevention of Torture and the League of Human Rights. The Belgian state was also repeatedly condemned by the European Court of Human Rights (ECHR).

Successive governments took different initiatives to address this problem. In 2012, mobile teams were created and six forensic networks covering the whole country were entrusted with the development of special individualized care pathways outside of prison psychiatric wards with the aim of reintegrating forensic patients into society. The objective was to integrate these care pathways into the emerging mental health care networks. To ensure the successful implementation of this initiative, two coordinators were hired for each network—one by the health sector and one by the justice sector, giving a total of twelve coordinators.

However, these plans did not succeed in solving the structural problem of the Belgian internment system. In 2016 the ECHR therefore issued a pilot judgment, once more condemning the detention conditions of forensic patients. In this context of international pressure the Prisons and Forensic Masterplan was jointly drafted by the federal ministries of health and justice and published at the end of 2016. This masterplan was presented as the continuation and improvement of past efforts to develop specialized care pathways to work alongside regular mental health care networks. The design of this plan followed a similar path to Innopsy107 and involved the same public officials and national experts. Judicial and forensic experts joined those who had created Innopsy107.

## 2.3. The Political Vision of Coordination

Both policy plans target complex policy issues that have not been resolved despite the existence of past policy initiatives. The issues are complex because they involve increasing cooperation between different sectors and services separated by institutional, organizational and professional boundaries. For example, the implementation of the Forensic Masterplan involves collaboration between the health and judicial sectors, both of which are from very different professional backgrounds—the former focused on protecting social order and the latter having the patient's interests at heart.

Furthermore, since Belgium is a federal state where responsibility for mental health and judicial matters is divided between several levels of power, hospital, community mental health, judicial and social services do not receive equal financial incentives to engage in policy implementation. Professional attitudes thus combine with institutional divisions to create boundaries likely to hinder policy implementation.

Finally, mental health care has developed in different ways in different regions—to reflect for example their urban/rural character—and the think tank knew that implementing the policy would mean adapting it to local circumstances. In the light of these circumstances, the think tank decided to devise a flexible framework for making room for the creativity of local stakeholders. It also made the network coordinators responsible for managing the local implementation of the model and issued the following description of their function:

First, the coordinator is responsible for mapping the existing care provision in his network area and for steering its development by setting up an evaluation system. He must also make connections

between different sectors such as the health, social and judicial sectors. Secondly, the coordinator actively participates in building a dynamic network: he must facilitate and gain the collaboration and trust of his network partners, build coalitions and create a learning environment fostering reflexivity. He is constantly focused on process improvement. Thirdly, the coordinator also contributes to the construction of care pathways and programs; he must pay attention to care quality and continuity and determine whether it meets the local population's needs. Fourthly, the coordinator ensures that his professional partners receive an adequate level of training to sustain the development of the network. In addition, the coordinator must be charismatic, patient, fair, empathetic and creative. He must have skills in situational leadership and have transdisciplinary, inter-organizational and inter-sectoral knowledge.

This definition illustrates the political vision underlying the function of coordinator. It originates in the design of Innopsy107 and was then replicated by other policy plans, including the Forensic Masterplan. In this description network coordinators play a strategic role in implementing both policy plans at the local level.

## 3. Theoretical Background: Policy Implementation as Translation and the Function of Coordination

The emergence of the function of coordinator and its associated challenges are closely related to an increase in the complexity of policy issues, which requires collaboration and communication among multiple stakeholders.

In addition, contemporary health policies are seeing a trend towards decentralization, which entails shifting power and accountability from the central state to regional and local networks of influence (Bannink and Ossewaarde 2012). Following this shift, local managers and professionals are made responsible for adapting policy plans to local realities; a process that often lead to renegotiating policy content and instruments with local stakeholders.

As a result, it is becoming difficult to distinguish between different stages of the policy process—i.e., agenda-setting, policy formulation, decision-making, policy implementation and evaluation (Howlett and Ramesh 2003). Indeed, one could argue that policy formulation continues throughout the implementation phase as those in charge seek to adapt policy plans to local realities. It could be equally argued that policy formulation and decision come before agenda-setting; national policy makers and experts may attend international conferences which then strongly influence the direction they take.

### 3.1. Policy Implementation as Translation

In this context, interpretivist approaches have called into question the concept of a linear transfer of policies. This linear approach assumes that policies are transferred across countries, governance levels and from one domain to another without changing meaning (Clarke et al. 2015, p. 13). Instead, interpretivist approaches emphasize the "unfinished" character of policy, suggesting they are reinvented during their implementation as a result of discussions and negotiations necessary to that implementation. New associations between different elements in the process (such as different stakeholders, ideas, interests and instruments) emerge as a result of this interactive process.

Assuming that policy changes as it "moves" means replacing the idea of transfer with that of translation (Freeman 2009), i.e., not merely a "simple transition but a selective and active process in which meanings are interpreted and reinterpreted to make them fit the new context" (Clarke et al. 2015, p. 35). Accordingly, in order to understand the outcomes of policy implementation, one should focus on the very practices through which meaning is recreated (Freeman et al. 2011).

In different policy fields, such as justice, health and education, intermediary actors are increasingly responsible for managing this translation process (Freeman and Sturdy 2014; Pichault and Schoenaers 2012; Radaelli and Sitton-Kent 2016). The network coordinators fall into this category. Therefore, this paper argues that in order to better understand how policy is implemented, i.e., "translated", the work of coordinators should be analyzed.

### 3.2. The Function of Coordination

Scientific literature conceptualizes coordinators as boundary spanners (Williams 2011). Boundary spanners include individuals who are explicitly responsible for connecting entities previously separated by a boundary (Kislov et al. 2017). It involves cooperation, better integration of services and efficient utilization of scarce resources (Williams 2002). Boundary spanners cope with their mission in four interrelated ways. First they develop, manage and sustain networks of relationships. Secondly they make things happen by being creative and seizing opportunities. Thirdly they manage relationships through appropriate communication fostering trust, listening and understanding and by translating between different professional languages. Fourthly, boundary spanners manage the development and maintenance of collaborative devices (Williams 2011).

The context in which boundary spanning occurs is uncertain and contradictory (Kislov et al. 2017; Marrone 2010; Williams 2002). It is characterized by the presence of diverse cultures, beliefs, interests and rigid institutional frameworks. There is often a lack of human, material and/or financial resources at the disposal of boundary spanners; at the very least these resources are unpredictable (Williams 2002, 2011). The context of healthcare is particularly challenging for those trying to work collaboratively by cutting across conventional boundaries (Currie and White 2012). Indeed, healthcare organizations are structured according to the model of professional bureaucracy (Mintzberg 1989), which keeps specialized domains of expertise and jurisdictions largely separate (Abbott 1988).

Moreover, the hierarchical position of coordinators has to be considered because it has consequences regarding their ability to perform boundary-spanning activities successfully. Coordinators occupy intermediary positions which provide them with resources but which also create obstacles when circulating new ideas across old organizational boundaries (Pichault and Schoenaers 2012; Radaelli and Sitton-Kent 2016). Being situated between senior managers and executives, they are "at once controller, controlled, resister and resisted" (Radaelli and Sitton-Kent 2016, p. 312). Occupying this position can be a hindrance, since coordinators cannot "fully rely on hierarchical position, resource control or expert knowledge to legitimize their involvement" (Radaelli and Sitton-Kent 2016, p. 319). Nevertheless, this position enables coordinators to access a wider range of knowledge (Rouleau 2005) and to mobilize other alternative sources of power such as pre-existing relationships in the organization.

To conclude, the concept of coordinators as boundary spanners highlights the importance of accessing a wide range of knowledge and translating between different professional languages and cultures. Translation induces different challenges. First, translation means transferring policy and professional ideas from one context to the next. Secondly it entails transformation, that is, shifting from ideas inscribed in documents to ideas embodied in instruments, by people, and enacted in social situations through verbal and non-verbal expressions (Freeman and Sturdy 2014). Thirdly, translation equally means new associations between previously separated ideas, objects and people, in a way that change their roles and relationships (Latour 1987).

How coordinators deal with these challenges is not questioned by the scientific literature on boundary spanning and on coordination. By addressing the question "what do coordinators do", this paper aims to identify practices used by coordinators when translating policy plans locally.

### 4. Materials and Methods

Our interest in the coordinators' work originated from two pieces of research focusing on the reform of mental healthcare (Innopsy107) and the creation of care pathways for forensic patients. Our intention is not, however, to showcase specific coordination work in either project. Instead we intend to illustrate practices and working conditions that apply to the function of coordination in both contexts.

Initially, the data collection was part of two research projects focusing more broadly on the implementation of Innopsy107 and Forensic. This exploratory stage allowed us to understand more generally the context in which the function of coordinator was emerging and evolving. Then, as our interest in the function of coordination increased, data collection focused on coordination itself.

This paper combines different methods of data collection including document analysis, semi-structured interviews and non-participatory observations. We suggest that this methodological triangulation (Jonsen and Jehn 2009) is required to understand coordination in a comprehensive way, by considering how objective representations and subjective experiences of coordination combine to shape the coordinators' work. We stopped collecting empirical data when we reached saturation of information (Kaufmann 2016).

We first carried out a comprehensive analysis of two main types of documents: regulatory documents, such as policy plans and legal frameworks, and organizational documents, including network agreements, project descriptions and annual activity reports.

In addition, we used semi-structured interviews to collect the discourses on coordination of coordinators and other stakeholders (cfr. Table 1 below). To assist us, we compiled an interview guide that was flexible enough to adjust to the trains of thought of the interviewees. This guide was designed to fit the stakeholder categories and evolved as the data was collected. Part of the interview guide dealt explicitly with the experience of coordination, but the interviews generally started by addressing the context the coordinators found themselves in. The interviews were always recorded and transcribed before being analyzed.

**Table 1.** Classification of interviewees according to their role in the implementation of the Innopsy107 and forensic care trajectories (CT).

| Classification of Interviewees According to Their Role in the Implementation of the R107 and Forensic Care Trajectories | Innopsy107 2010–2017 | Forensic CT 2016–2017 |
|---|---|---|
| Policy makers and Coordinators responsible for supervising policy implementation | N = 13 | N = 4 |
| Coordinators responsible for loco-regional implementation | N = 13 | N = 12 |
| Coordinators responsible for psychiatric mobile teams | N = 5 | N = 4 |
| Field actors including justice/health professionals and psychiatric hospital managers | N = 13 | N = 18 |

In particular, the coordinators we met have varied profiles in certain respects, for example in terms of age or seniority. Although most of them are psychologists or social workers by training, some of them already held "executive", coordination or management positions before being recruited as coordinators.

Finally, we used direct and nonparticipant observation to see coordination in action, i.e., how coordinators enact their work through meetings (cfr. Table 2 below). Observing meetings enabled us to understand how coordinators circulate information, build agreements and guide local professionals in their understanding of mental health and forensic policy plans. In looking at these aspects of the coordinators' work we paid particular attention to how participants contributed to specific meeting discussions. Observation (Peneff 2009) of meetings was never recorded. However, systematic note-taking of certain conversations as well as the context and general course of the meetings allowed us to easily access empirical material.

**Table 2.** Classification of meetings according to decision-making level and the type of professionals involved.

| Types of Meetings | Innopsy107 2010–2017 | Forensic 2016–2017 |
|---|---|---|
| Policy and strategy meeting: meetings between policy makers, public health authorities and local coordinators | N = 3 | N = 6 |
| Network or steering committee meeting: meetings between managers of mental healthcare services and institutions | N = 16 | N = 6 |
| Working committee meeting: meetings between frontline professionals | N = 27 | N = 0 |

We conducted a thematic content analysis (Paillé and Mucchielli 2012) for the different types of data collected. Following grounded theory methodology (Morse et al. 2009), the analysis categories were gradually drawn from our empirical observations. These categories allowed us to order and

analyze our data systematically. Documents, transcripts of interviews and observation notes were therefore subjected to several rounds of reading and analysis. The analysis categories could then be developed and structured to create the arguments set out in this paper. To protect anonymity all names used in this manuscript are pseudonyms.

To conclude, (i) our continuous and long-term presence in the field, (ii) the observation of meetings gathering rather stable groups of people, (iii) the interviews of those actors and (iv) the systematic comparison of different types of data allowed us to follow the translation process as well as to see stakeholders' representations evolve as policy ideas evolve.

## 5. Results

With this section we aim to reflect the day-to-day reality of how coordination is practiced. First, using interviews, observations and document analysis, it describes the coordinators' uncertain working environment and empirically illustrates how the definition of the function is socially disputed. Secondly, it asks how coordinators perceive themselves and suggests that they enact a sense of who they are by talking to other stakeholders and to each other about the function of coordinator. Thirdly, it examines the two core practices of coordination work: meeting and working with documents.

### 5.1. The Coordinators' Working Context: Novelty and Uncertainty

Coordination is a new and widespread phenomenon. Many stakeholders in the mental healthcare and forensic systems have emphasized the significance of this new type of professional: *"New kinds of jobs appeared with the reform, as did the coordinators ... Coordination is becoming increasingly necessary. There are lots of new challenges ... it is all very complicated." (Coordinator, interview, 2016/06).* There are different types of coordinators and they work in different arenas: *"When we talk about coordinators who do we actually mean? Those working in the legal sector? In public health? We also have coordinators within the hospital itself. It's getting out of hand, a little messy." (Director of a psychiatric hospital unit, interview 2016/06).* Despite this variety of forms of coordinator they are all entrusted with a political mandate to transpose federal policy into empirical reality: *"To a large extent, the success of the reform depends on the coordinators ... They have to build integrated networks with clear processes and procedures ... The coordinator is the driver of the process." (Federal coordinator, interview, 2012/03).*

Coordinators perform this challenging task in an ambiguous context. The role of partner organizations in developing local networks is always unpredictable: these organizations preserve their autonomy and may therefore pull out at any time. The only way for coordinators to secure partner organizations for a period of time is to sign collaboration agreements, which sometimes have to be renewed every year. The composition of the coordinators' networks is highly dependent on whether the partner organizations are committed to the ongoing reforms.

At the same time the coordinators' position within their developing networks depends on political decisions outside the coordinators' control:

> *"We have been working for two years now. Our teams are highly motivated but our project still hasn't been given the go-ahead [by the policy makers]!" (Coordinator, local meeting, 2012/02).* Another coordinator told his network partners: *"It is uncomfortable to have to say, 'follow me, we're creating a network ... but not just yet (because my project is yet to be approved)'." (Same meeting, 2012/02).*

Finally, coordinators appear to be trapped between the political and organizational worlds, with a need to reconcile different visions, interests and different working rhythms. They exert no control over what happens in these two worlds despite their impact on the development of their networks.

### 5.2. The Function of Coordination: A Disputed Definition

The function of coordination is socially disputed. Stakeholders interpret the coordinators' mandates in various ways and contest each other's perceptions. The main ambiguity lies in the gap

between the policy makers' concept of coordination, which is viewed as embodying policy objectives, and that of professionals in the field.

Their discourses enact noticeably different meanings of coordination. Firstly, the work of coordinators is often perceived as less visible and not as practical or useful as the patient-centered work performed by field professionals: *"I do not want coordinators, I want people who work." (Federal advisor of the Health Cabinet, interview, 2017/04)*. Secondly, faced with the rapid growth of this new and widespread function, the stakeholders regularly experience coordination as a threat: *"It is worrying that increasingly, there are more coordinators than health professionals" (frontline professional, local meeting, 2011/11)*. Thirdly, coordination is frequently depicted as a frustrating, unrewarding job, both because the working environment is unpredictable and because of the nature of the coordinator as intermediary. *"It's a thankless task. If the project fails it's the coordinator who gets the sack!" (Manager of a community mental health service, interview, 2012/02)*. Coordinators are indeed the target when there is criticism of policy programs and the way they are implemented: *"It is the coordinator who gets the brunt of the criticism. You need a thick skin and the ability to distance yourself from what people say." (Coordinator, interview, 2017/06)*.

By contrast, policy makers and civil servants share a clear vision of the coordinators' mandate: *"For the federal administration the network coordinator is the main person of reference: all our requests pass through the coordinator." (Federal coordinator, local meeting, 2012/12)*. This strategic concept of coordination provokes strong reactions from field professionals: *"Public authorities put the coordinators between themselves and local care providers. Coordinators are in a very vulnerable position with regard to public authorities, and these authorities use different forms of coordinators to communicate decisions to local care providers." (Manager of a psychiatric hospital, local meeting, 2014/01)*. One coordinator explained: *"In the past institutions communicated directly with the federal level [...] whereas now the federal authorities are trying to put a stop to this by stating that if there is a request, it has to pass through the coordinator. The coordinator acts as a kind of filter; although the federal authorities would never put it in those terms, that is precisely what they want the coordinators to be ... " (Coordinator, interview, 2017/06)*.

The stakeholders' discourse on what coordination is reveals not only its disputed nature but also its significance for field professionals; for them, the proliferation of the coordinator is a symptom of a new division of work and new power relations within the mental health care and forensic systems.

*5.3. Beyond the Political Vision of Coordination: Coordinators as Connectors*

The political vision of coordination presented at the outset of this paper appears unrealistic and too abstract to coordinators: *"If you have a look at the job description you'd have to be superman to have all those skills... and yet you still don't really know what is expected of you." (Coordinator, interview, 2017/08)*. Another coordinator explained: *"At first no, you don't know what you'll be doing nor how to tackle the job. We're talking about coordination at the federal level; it's all rather vague and mysterious." (Coordinator, interview, 2017/06)*.

Without a practical description of their function, how do coordinators make sense of their work? The coordinators enact a sense of who they are, namely connectors, by talking to other stakeholders and to each other. Every act of coordination, usually occurring during inter-organizational meetings, refines and complements the coordinators' understanding of their work.

When asked, "What is a coordinator?" our interviewees first listed the activities in their formal mandate, such as recruiting new networks and promoting care for forensic patients. Then, when asked how they specifically performed these activities, most of them eventually concluded that: *"the main activity and often the only job of the coordinator is to create links between institutions and bring added value to the existing network." (Coordinator, meeting, 2011/11)*.

Moreover, according to our interviewees, making links might be the only way for coordinators to exert any power—a conclusion they reached by discussing the subject with each other: *"There are two of us, Maurice, and we have absolutely no power!" Maurice answers: "but at least we have the power to make connections." (Interaction between two coordinators, meeting, 2012/12)*. This interpretation was echoed by field professionals:

*"You said power!? No, they have no power at all. The power belongs to the care institutions and their role is to streamline inter-structural contacts. But perhaps that is still a kind of power, even though they have no decision-making power as such. Instead they can initiate, propose, advise, listen . . . and I would say . . . make connections." (Psychiatrist, interview, 2011/12)*

Unlike hierarchical power or authority, the power to make connections applies to processes rather than workers: *"Coordinators exercise power over processes, not individuals - never. However you do have power over the 'hows': you manage meetings and how they turn out; you take the lead." (Federal coordinator, interview, 2017/08).*

Coordinators therefore make sense of their function and uncertain work environment through discussion. It is indeed through talking to us and to each other that they came to see themselves as connection-makers, deriving power from their ability to link things together; in so doing they drive policy implementation.

### 5.4. Meetings and Interpreting Documents

A close look at the coordinators' work during the two above-mentioned research projects over a period of eight years has led us to the conclusion that coordinators spend most of their time either in meetings or writing documents.

Meetings are where coordinators handle the wealth of knowledge available in their environment. They use them to circulate policy objectives and collect information on partner organizations in the networks: *"I meet the stakeholders, I give them information and I find out what already exists" (coordinator, interview, 2012/01).*

During meetings, coordinators establish connections between previously separate, unrelated elements. Combining experience and contextual information in new ways with the cooperation of those attending the meeting is their way of making sense of the complex policy processes they are engaged in.

In the following examples we show how coordinators connect ongoing discussions during meetings with issues that are often separated in time and space. This may include documents, past experiences, discussions and meetings that may have occurred elsewhere or in a different context.

Firstly, to prevent traditional professional culture invading the meeting room, a coordinator's strategy is to recall the philosophy underlying the policy plan: *"We cannot continue to think in terms of 'he is the patient of so-and-so' but instead we should approach this in terms of functions. The whole point [of the policy plan] is to activate functions." (Coordinator, local meeting, 2012/02).* In doing so, the coordinator connects the ongoing discussion with the vision set out in the policy plan.

Secondly, coordinators attempt to redefine the situation by combining differing interpretations of the same phenomenon, for example 'crisis situations': *"On the subject of beds in a crisis, what do we actually mean by 'crisis'? It is essential we mean the same thing if we want to create something together." (Coordinator, local meeting, 2017/06).* In this way the coordinator tries to create a common sense of "crisis situations" among the professionals at the meeting.

Thirdly, creating links between past policy initiatives and ongoing discussions is another technique used by the coordinators as a way of facilitating discussions at meetings. In the following illustration, past experiences are recalled by the coordinator with the aim of both playing down the participants' reaction to policy change and of reassuring them that they hold the necessary expertise to make the change a success:

*"This reminds me of the issues that arose when implementing the therapeutic projects [pilot projects which started in 2005]; those involved in the projects benefited in terms of knowledge, interconnections and coordination." (Coordinator, local meeting, 2012/01).*

Fourthly, another strategy used by coordinators to give a particular direction to the interpretation of a document is to link it to another, initially independent document: *"That is a political document, but I think that their intention is to promote therapeutic consultation for psychiatric patients. If you look at the*

*document of the National Institute for Health and Disability Insurance, the similarity becomes quite clear."* *(Coordinator, local meeting, 2010/01).*

Finally, the coordinator can also attempt to define the situation by linking ongoing discussions to other discussions or events, separated in time and space. The two examples below show that ongoing discussions are embedded in networks of events and meetings and that coordinators can give sense to those ongoing discussions by referring to both past and future events. In the following example the coordinator refers to a very recent meeting in order to place the discussion in context:

> *"Only yesterday Mister X [senior officer at the federal administration] recalled the fact that in the reform project at federal level, the third care function [= rehabilitation, the focus of the meeting] is intended to link care to other types of services . . . " (Coordinator, local meeting, 2012/02).*

Then, by referring to a future meeting, the coordinator gives meaning to the discussions: *"That is the kind of tool the federal leaders of the reform are interested in. We have a meeting at the federal administration on Friday to learn about good practice relating to this instrument." (Coordinator, local meeting, 2012/02).* In this way the coordinator reminds people of the vision and provides a strategic direction for the ongoing discussion at the meeting.

It follows that coordinators enact their work by reconnecting their world. Creating networks of events, including meetings, is one of the core practices of coordination. It allows coordinators to attach sense and substance to ongoing processes that cut across multiple arenas and unfold in an experimental and uncertain framework. We suggest referring to this practice as "event connectivity" (Thunus et al. 2019), which emphasizes how, by linking originally separate events, the coordinators enact one of the possible futures of either policy plan.

*5.5. Creating Documents*

As mentioned above, working with documents is also characteristic of the work of coordinators. Our empirical material allows us to illustrate how coordinators use documents as part of an active strategy to frame and give shape to meetings but also to secure the meaning that has been enacted through meeting conversations.

In the following example the coordinator initially asked the meeting participants to reflect on a local definition of a certain care function outlined in the policy plan, i.e., mental health promotion and prevention. The ensuing discussion illustrates how the coordinator managed the local "translation" of the policy document by "sifting" (Weick 2015) through the content of the discussion and gradually moving from a collective and verbal form of (enacted) knowledge to a written or 'inscribed' one. The coordinator's challenge is to select relevant pieces of information and create a common definition reflecting the participants' viewpoints. This conversation demonstrates the contingency of the policy plan meaning enacted at meetings:

> [13]: *"We must emphasize the idea of multidisciplinary work".*

> [14]: *"Intersectoral work as well".*

> [15]: *"I disagree. The term intersectoral includes the justice system too. That would raise issues such as medical confidentiality, ethics and so on. The intersectoral approach belongs to the fields of care and assistance".*

> [16]: *"I think that the idea of 'shared values' could help us going forward".*

> [17]: *"Would not it be better to start with a few introductions—get to know each other, find out our strengths and where our limits are . . . "*

> [18]: *The coordinator responded: "It would be time-consuming".*

> [19]: *"Our knowledge will become more visible as we interact. This will be a gradual process. It will be interesting; gradual but continual progress".*

> [20]: *"Don't we have to decide what the outcome should be? For instance: the interests of the patient?"*

Following this conversation, the coordinator put the ideas of the participants into the following categories: (1) Active accessibility, (2) The WHO definition, (3) Primary care (close to home), (4) Networking, (5) Care and assistance, (6) Multidisciplinary work and (7) The interests of the individual. He proposed drawing on these categories to define the general objective: "Partners in care function number 1 have the following goal ... ".

The coordinator's intervention consists in selecting elements enacted by the participants and in rearticulating them by substituting a logical order to the chronological order of the participants' speeches. These processes of selection and articulation—which we propose to refer to as "contextualizing"—are central to the creative process underlying policy translation. However, it must be noted that the participants' interventions are always partly unpredictable. The coordinator can never predict how each person is going to respond to a question or react to an intervention. Therefore coordination appears to be largely a process of translation, the result of which is secured through documents.

## 6. Discussion

The results largely endorsed scientific literature, which emphasizes the scarcity of resources and the uncertainty of the environment in which coordination is usually performed. In addition, they illustrated constant disagreements among stakeholders about the relevance and scope of the function of coordinator, making it a function subject to dispute. This highly contested trait is a serious obstacle for coordinators who are expected to cross professional and organizational boundaries.

Moreover, the results bear out the claim that coordinators have no formal authority over professional stakeholders. They reintroduced the distinction in scientific literature between authority based on hierarchical positions and that of coordinators, which is derived from their access to a large number of stakeholders and broad body of knowledge across different sectors and organizations. Indeed, the transcript of a discussion between two coordinators highlighted the fact that in their view the power of coordinators lies in their capacity to make links between separated elements. This power can then be applied to a process rather than to people and structures.

Based on these results, this paper suggests redefining coordinators as a particular kind of governance practitioner managing policy processes in ambiguous environments (Weick 1998). Indeed, according to Weick, defining features of ambiguous situations are an uncertain environment, a lack of resources and complex, changing problems with unknown or contested solutions (Weick 1995). Furthermore, managing processes in such environments can be described as mainly consisting of improvisation, which "involves reworking precomposed material and designs in relation to unanticipated ideas conceived, shaped, and transformed under the special conditions of performance, thereby adding unique features to every creation" (Weick 1998, p. 241, cited in Weick 1998). Improvisation can thus be viewed as a situated instance of the process of translation, which ongoingly gives shape and content to policy plans. Its outcomes are unpredictable and unique—both personal and social creations. In the remainder of this discussion, we will refer to this notion of improvisation to draw lessons from the analysis of our empirical material.

Our analysis drew attention to meetings, which are the setting where coordination takes place, as well as documents that emerged as the preferred method of coordinators for 'securing' each of their creations. Moreover, this analysis shed light on "the reworking of precomposed material", i.e., policy plans and related documents, and emphasized how this reworking operates.

Making links appears to be a common step taken by coordinators to achieve the local translation of both policy plans. Firstly, coordinators connect practical concerns in a given meeting to abstract policy objectives. Secondly, they link political and professional concerns. Thirdly, they combine events that were previously separated in time and space. Fourthly, they draw parallels between policy plans and other documents issued in different contexts and for different purposes. Linking incommensurable events, a process we suggested referring to as "event connectivity" (Thunus et al. 2019), might therefore be an important characteristic of coordination performance: it is the way in which they actually make things happen.

The type of resources used by coordinators when trying to reconnect segmented worlds is also evident in the way they approach meetings. These resources include their personal history and involvement in multiple events and networks. Improvisation is therefore strongly grounded in identity construction and the principle of continuity of experience—implying that "every experience modifies the experiencer" (Weick 2015, p. 121), appears central to understanding coordinators' resources to translate policy plans locally (Weick 2015). Indeed, the more varied someone's past experience is, the more sources of inspiration there are which can be applied to the situation: "the more selves I have access to, the more meanings I should be able to extract and impose in any situation" (Weick 1995, p. 24).

However, personal experience and involvement in multiple activities are not the only resources required to perform coordination work. Improvisation is ongoing and social; it occurs in public. This public—the meeting participants—takes an active part in coordination performances by responding to coordinators' interventions and by commenting on various documents. Moreover, participants' contributions depend on how they perceive the situation and are thus highly unpredictable. Thus, coordinators have to be able to understand and interpret what the meeting participants mean when they describe something, before fitting their contributions to what the group as a whole has created.

Our observations of meetings chaired by network coordinators showed that they select specific contributions irrespective of when they were made. They then connect them to the objectives of the policy in question. This operation that we described as contextualizing (Corbin and Strauss 1993; Strauss 1988) consists in placing what has been said in a logical rather than a chronological order. This, in our view, is how coordinators face the uncertainty surrounding improvisation and how they link the resulting creation to policy objectives.

At the end of this discussion, three main lessons can be learnt from our empirical analysis.

Firstly, the disputed nature of the function of coordinator is a serious obstacle to coordinators achieving their objectives. In order to create connections between different sectors and organizations, coordinators have to appear credible to the professional stakeholders. However, as we have shown, these stakeholders deplore the proliferation of coordinators, interpret their role differently, question the utility of this function and sometimes see coordinators as manipulated by policy makers. In this context, this paper suggests that with a view to decreasing the ambiguity surrounding coordinators, their mandate could be clarified and refined in collaboration with different stakeholders. This endeavor could lead to reviewing the description of the function in a way that considers practical experiences of coordination, including the importance of improvisation.

Secondly, coordination performances do influence the meaning that policy plans acquire throughout the implementation phase. As suggested in this paper, this phase has emerged as a translation process through which the content of policy plans is constantly improvised. Moreover, since improvisation is embedded in continuous social identity construction, it is largely unpredictable.

Consequently, the recruitment and training process of coordinators is not appropriate for their work. It is based on the political vision that sees the function as that of an intermediary transferring the content of policy plans without changing it. However, this paper stresses the significance of event connectivity and "contextualizing", two methods via which coordinators improvise the meaning and scope of policy plans.

Attaching importance to improvisation would refine recruitment and training in two ways. Firstly, greater attention could be paid to the resources that coordinators use to improvise. We would suggest that the more experienced, flexible and mobile coordinators are, the more sources of inspiration they have and therefore the more relevant their improvisations are likely to be. In fact, experience, inter-sectoral and international mobility provide coordinators with ideas they can mobilize for improvising. Secondly, there is room for improving the skills of coordinators when they organize meetings, select participants and report on the discussions at those meetings. Indeed, our results emphasized the extent to which not only the discussions but also the contributions of every meeting participant support the improvisation process.

Thirdly, viewing coordination as improvisation has important implications for the steering and evaluation of policy implementation. Indeed, steering and evaluation are currently procedural and based on command and control mechanisms. In other words, policy makers attempt to control the process by checking the compliance of coordinators with procedural requirements, such as the obligation to organize a fixed number of committee meetings per year. Moreover, the annual report issued by coordinators focuses more on quantitative indicators, such as the number of services involved in network construction, than on creative adaptation which influences the meaning of policy plans. Consequently, policy makers are usually unfamiliar with the extent to which the meaning of policy plans changes at the local level and have an inaccurate idea of the progress and outcome of policy implementation. Evaluation could be more qualitative, reflective and adaptive, that is, based on discussions enabling reciprocal adaptations in both the policy plans and their local translation.

## 7. Conclusions

This paper questioned the work of coordinators and their influence on the local translation of policy plans. These plans involve complex policy issues and endeavor to convey international policy ideas on how to foster the integration of people with mental health problems into society.

By drawing on qualitative material, this paper suggests seeing network coordinators as a particular kind of governance practitioner whose work mainly consists in translating the meaning and scope of policy plans locally. Meetings have emerged as the setting for improvisation; documents are then the means through which the result of that improvisation takes on practical form. In addition, based on in-depth analysis of direct observations of meetings, this paper highlights two operations through which coordinators improvise the meaning and scope of policy plans, namely, event-connectivity and contextualizing.

This empirically-grounded definition does not corroborate the political vision of coordination, which describes coordinators as simple intermediaries implementing policy plans locally, without changing them. Instead, this definition emphasizes the active contribution of coordinators to a process that is best defined as a 'translation'. This paper therefore suggests paying more attention to the importance and implications of improvisation on two levels: rethinking the recruitment and training of coordinators and the steering and evaluation of policy implementation. Finally, with a view to decreasing the ambiguity surrounding coordinators, their mandate could be clarified and refined in collaboration with different stakeholders. This could lead to revising the description of the function in a way that considers practical experience of coordination, including the importance of improvisation.

The limited scope of the two research projects which led to this paper as well as the use of qualitative methods are likely to restrict wider application of the key lessons and conclusions drawn about coordination work. However, the importance of network coordinators to the implementation of social and health policies is growing in OECD countries. The conclusions of this paper could therefore assist further practical as well as scientific debates on function of coordination.

**Author Contributions:** Both authors contributed to the different steps of the article conception and writing. All authors have read and agreed to the published version of the manuscript.

**Funding:** This paper derives from two research projects that have been carried out thanks to the support of the Belgian National Fund for Scientific Research (F.R.S.–FNRS) and the Federal Public Service Health, Food Chain Safety and Environment (For-Care Research). We thank the F.R.S.–FNRS and the Federal Public Service Health for their financial support and for preserving the scientific independence of the research process.

**Acknowledgments:** We are particularly grateful to Richard Freeman and Natalie Papanastasiou for having read and commented on the previous versions of this paper. We thank ECPR for giving us the opportunity to present a first version of the paper during the 2017 General conference in Oslo. We also thank Frederic Schoenaers for his continuing support to both research projects, and our colleagues from the Sociological Research and Intervention Centre (CRIS) for our discussions on the coordination function in contemporary health policies.

**Conflicts of Interest:** The authors declare no conflict of interest.

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
