# Peer review of "What Do Coordinators Do? Mental Health Policy Implementation as Translation"

_admsci, doi:10.3390/admsci10010009_

Round 1

Reviewer 1 Report

This paper is a valuable study of coordination in and as practice, situated, embodied and interactional.  It draws on a significant volume of data, and has both an articulate/theorised sense of coordination work as process and some useful practical implications.  

'Translation' is the paper's key term, and I suggest it replace 'improvisation' in the title of the paper.

Having explained that the coordinator is a particular kind of governance practitioner, the introduction might ask a bit more simply somewhere: 'What do coordinators do?'

Both here (introduction) and 158ff below: I wonder whether the coordination/translation function is specific to international policy?  Cross-national transfers are interesting because they make translation an immediate and visible problem, but translation is always going on, even among practitioners and organizations who think they know each other well.  We gave to translate from health care to social care, for example, as well as between natural languages.  Cross-national transfer prompts the investigation/discussion carried by this paper, but they're more widely applicable.  It would be more interesting and compelling I think, to speak of translation as a general effect of network governance (which assumes/requires communication among multiple different actors).

119ff the political vision of coordination is well expressed.

Section 3, 154ff: theoretical discussion is of the boundary spanner, who I would treat more briefly: explain that the boundary spanner is also and almost by definition a translator - and then focus more clearly on the work of translation.

Section 4, 249ff methods: what did the research team do to follow translation as a process, from one meeting to the next, one document to another?

297ff is misnumbered Section 3

3.4 the point that connections are made in meetings, and the importance of connecting across meetings, is interesting/important and well made

3.5 the treatment of documents offers no real evidence of a coordinator drafting or producing a document, though surely it should - cf idea of 'securing' creations 521-2 below

4. Discussion of making links 524 is again important and well done.

Author Response

Dear reviewer,

First of all thanks for the relevant and interesting review. Here is our answer :

Improvisation has been replaced in the title and throughout the paper. It was our initial intention to use translation as a central concept but we changed it following a previous reviewing process. So thanks! Same remark applied for the expression “what do coordinators do”. This question has been integrated to the core of the paper We agree that the need for translation is not specific to international policy but is required by network governance. However, international organisations played significant influence on both policies analysed through this paper. Therefore, the stress put on the international dimension refers to the context rather than to the specificity of the coordination function. Consequently, we kept the reference to this dimensions within the context section but deleted it from the theoretical section. Regarding section 3, we now focus more clearly on the work of translation and the question of boundary spanning is consequently less developed Regarding methodological section, we explain it now more clearly how methodology helped us to follow translation as a process at the end of the section Thanks, we renumbered the section and following ones Regarding the coordinators’ use of documents, we reorganised the section to shed light on empirical evidence illustrating this.

Thanks again ! The changes brought following the two reviews are yellow highlighted within the text.

Reviewer 2 Report

This paper presents a well-structured, clearly written analysis of some of the features and challenges of the function of « coordination » regarding two fieldwork dealing with two mental health-related policies.

The developments are clearly presented and the rationale is easy to follow. The readers gets the information they need (or most of it, cf infra) to understand the complexity of the (belgian) policies context.

Here follow some comments, which are not to be considered as critics but as points of discussion/questions, which could be taken into account and implemented in some minor revisions.

I think the only important information that is missing concerns who the coordinators really are. How have they been recruited ? What where their previous job ? Did their compliance with these policies implementation play a role ? If coordinators are mainly former caregivers, this is an important point to understand why they are led to some kind of improvisation.

127 : I wonder if reducing the consequences of the division (or you could say dissemination) of responsibilities in various levels of power in Belgium to differences in financial incentives is not a bit simplisitc. Of course, this aspect is of the utmost importance. But there are probably other elements such as the political culture or, regarding the mental health area, an objective context (such as the number of psychiatric beds, the theoretical culture) which leads to differences in interest, perceived gains in such policy implementation, and thus differences of attitudes towards it. The sentence 132 announces this but does not develop the point. I think it would be interesting to give more detail to show better the tension between general guidelines supposed to be applicable everywhere and local adaptation expected from coordinators.

138 : this paragraph is very interesting. The reader may be interested to know with which material this « ideal-type » of the coordinator has been sketched. Moreover, the sentence 150 say that this definition illustrates the political vision underlying… » but the reader expected the author(s) to sum up what is this political vision.

In this point too (or could be dealt with in the following point 3 about theoretical background) : I think that another feature of the context which is needed to understand why these policies rely on coordinators is the general success for some decades now of the « network » category in policy implementation. This category has been hailed as a solution to the problem of the « institution » and its supposed rigidity (and there is here an interesting parallel between the critique of the institution as an old fashioned way to implement policies and, on the other side, in the mental health field, the critique of institution as a disrespectful and unefficient way of treating patients/clients).

I also link this success to the rise of attention given to the processes, the way individuals, groups or institution do communicate, and the ways this communication could be improved (a common topic in self-help books for managers for example). In short, this trend leads to a focus on procedural issues (especially when things are complicated) and the hypothetical potential for improvement.

192 : As explained in the literature review, the focus on the way various actors translate (instead of transfer) policies is quite well-known in the literature nowadays. However, my impression is that sociologists have been focusing on the room for manoeuver of first-line actors (for example in social service) to show the way they interpret and enact the guidelines that are imposed on them. But the function of coordinator under scrutiny here is different, and I think it could be interesting, in order to underscore the novelty of the analysis here to say a word about the potential difference between previous analyses on first-line actors, and coordinators (the answer is already in the paper, with the notion of boundary spanner)

201 and sq : the sentences are very affirmative (for ex. 206). Is it the case in any context ? Maybe adding a « maybe » would be more careful

The sentence line 240 is important and maybe need some more developments : how and why are coordinators not being associated with ideologies and strategies ?

Methodo : line 268. The reader is interested in how coordinators have been recruited for the research. Has there been some refusals ?

Line 289 : as there is a quite improtant quantity of material, a little bit more could be said about the way the grounded theory was applied and led to these results. If available, the categories grid could be annexed to the paper.

Results :

This section is very interesting. What was not clear to me was the level of knowledge and sympathy (or antipathy) stakeholders have for the two policies. I think it is an important feature of the context in which coordinators operate. Even before they implement anything, do they have to inform, explain or convince stakeholders (which might think the reform is only financially motivated, for example)? I think it is interesting as Innopsy seem to expect from the coordinators actions on the processes and the practices, but that the reality of the coordinators’ job is to deal mainly with representations, idea(l)s, symbols,…

Line 380 sq. To understand the power (or the absence of it) of coordinators, one should also be informed of the way financial resources are distributed in the mental health sector (to hospitals,…). Also : the description of connecting people as a source of power is interesting but quite counter intuitive in strategic analysis, where power is often understood to come from disconnecting people (structural holes, marginal sécant,…°

Discussion :

I was wondering if the notion of improvisation, which is interesting, might though sound  as pejorative and might induce the idea that coordinator do not follow any rules (which is not what the paper suggests). In recent work about the way psychiatric emergency reinvent solutions for each patient and situation, I have used the idea of an « artisanal » aspect of jobs that face so many contingencies.

Last comments :

Line 339. What is a federal advisor ? (not to be confused with federal coordinator)

I was not sure about the meaning of the sentence beginning line 116. A last check about editing (full stops, commas and italics) may be needed.

Author Response

Dear reviewer,

First of all thanks for the relevant and interesting review. Here is our answer:

We added in the methodological section a brief explanation on the coordinators’ profiles and backgrounds. However, the need for improvising should not be understood as a consequence of their background and professional trajectories but in relation to the uncertainty of their working environment and the problems they are facing. We mention the existence of various professional cultures in the previous paragraph. We now specify that prof attitudes and sectoral cultures combine to hinder policy implementation, next to institutional divisions. The “ideal type” of the coordinator: this definition is a summary of different information collected from organisational and policy documents describing the function. This is why it reflects the political vision of the function. Thank you for this very interesting comment. It could be relevant to reflect on the withdrawal of institutions from both implementing policies and treating patients. Indeed, at these two levels, the withdrawal of institutions seems to make room for the development of new concepts, ideologies, professional functions …which have some similarities. One of them is to displace the responsibilities for making sense of the situation from institutions and the rules they embodied to individuals (either patient or practitioners). This idea could be deepened in another paper, but it is not really needed, in our view, to grasp this papers’ argument. Regarding the added value of the research, thanks, we added it at the end of the theoretical section. We followed that advice and softened the affirmative side of that sentence. We shortened some aspects of the theoretical section so this idea do not appear anymore as it was. However, the idea is approached just in the previous paragraph. All coordinators have been met, no refusal, this is why we do not address this issue in our methodology. Regarding the level of knowledge and sympathy of stakeholders: it is true that coordinators have to interests stakeholders but we understand this as part of implementation through translation practices and not as a previous stage. Regarding the distribution of financial resources, we affirm in the methods section that we do not wish to wish to address the specificity of the function. This funding method is specific the mental health field and does not apply to forensic coordinators. Following the advice of another reviewer we now use simply the idea of translation rather than improvisation within the title and in some other places of the paper. However, we decided to keep the notion of improvisation since it sheds light on coordinators’ creativity and ability to deal with ambiguity rather than giving a pejorative account of their performance. In our view, the idea of improvisation encompasses to some extent that of “artisanal” in the sense it underlines the unpredictable aspect. We speak about the federal advisor of the Cabinet, which is not precised. Thanks!

Thanks again. The changes brought following the two reviews are yellow highlighted within the text.